# The "Self" under COVID-19: Social role disruptions, self-authenticity and present-focused coping

**Jingshi (Joyce) Liu[1], Amy N. Dalton[2], Jeremy Lee[2]\***

**1** Faculty of Management, Bayes Business School (formerly Cass), City, University of London, London, United Kingdom, **2** Marketing Department, School of Business and Management, Hong Kong University of Science and Technology, Hong Kong, Hong Kong

\* yxjlee@connect.ust.hk

**Data Availability Statement:** All relevant data are within the manuscript and its Supporting Information files. The dataset is stored in OSF with the following URL: https://osf.io/f6abv/?view_only=

## Abstract

Social role disruption is a state involving upheaval of social identities, routines and responsibilities. Such disruption is presently occurring at a global scale due to the COVID-19 pandemic, which poses a threat not only to health and security but also to the social roles that underlie people's daily lives. Our collective response to combat the virus entails, for example, parents homeschooling children, friends socializing online, and employees working from home. While these collective efforts serve the greater good, people's social roles now lack continuity from what was authentic to the roles before the pandemic began. This, we argue, takes a psychological toll. Individuals feel *inauthentic*, or alienated and out-of-touch from their "true" selves, to the extent their social roles undergo change. As evidence, we report survey (Studies 1 & 4) and experimental (Studies 2 & 3) evidence that COVID-19-related role changes indeed increase inauthenticity. This effect occurs independent of (a) how positively/negatively people feel about COVID-19 (Study 2) and (b) how positively/negatively people feel about the role change itself (Studies 3 & 4). Moreover, we identify two moderators of this effect. First, this effect occurs when (and ostensibly because) the social roles undergoing change are central to an individual's sense of self (Study 2). Second, this effect depends on an individual's temporal perspective. People can safeguard their self-authenticity in the face of changing social roles if they stay focused on the here-and-now (the present and immediate future), rather than focusing on the past (pre-COVID-19) or future (post-COVID-19) (Studies 3 & 4). This advantage for present-focused coping is observed in both the U.S.A. (Study 3) and Hong Kong (Study 4). We suggest that the reason people feel more authentically themselves when they maintain a present focus is because doing so makes the discontinuity of their social roles less salient.

## Introduction

Social roles are fundamental to people's sense of self. Accordingly, we define ourselves in relation to our social roles [1]. However, as the circumstances around us change, so too do our

5bf2b36dc9d448f2a1638ab698c9a28f; DOI: 10.
17605/OSF.IO/F6ABV.

**Funding:** This research received financial support
from Research Grants Council (RGC) grant RGC-
GRF-16501420 awarded to AND. https://www.ugc.
edu.hk/eng/rgc/ The funders had no role in study
design, data collection and analysis, decision to
publish, or preparation of the manuscript.

**Competing interests:** The authors have declared
that no competing interests exist.

social roles. The recent coronavirus (COVID-19) pandemic is precipitating change at a global
scale, and collective efforts to combat its spread have led to profound and unprecedented dis-
ruptions to social systems and roles. As of August 2021, global COVID-19 cases continue to
rise and social systems remain in flux and disarray, with nearly 60% of the world population
subject to some form of restriction in movement [2]. Meanwhile, as the pandemic sweeps
across the world, various populations are encountering new and renewed disruptions. In
India, the government implemented a fresh series of stringent regional lockdowns in response
to the record-breaking spike in daily new cases in late April [3], while Australia has been
renewing regional lockdowns throughout the summer months [4].

Although collective action to stop the spread of the pandemic is critical, the lives of individ-
uals who function within these social systems are upended. A person navigating social roles as
a parent, employee, and friend, for example, now faces different responsibilities (e.g., home-
schooling children), behavioral routines (e.g., wearing a mask in the workplace), contexts for
role execution (e.g., socializing online), or even suspension of roles entirely (e.g., furloughed
employment). Social role changes under COVID-19 are noteworthy for the threat they pose to
people's interpersonal relationships and psychological wellbeing. For example, parenting dur-
ing lockdowns heightens parent-child conflict [5] and produces emotional exhaustion, stress,
and burnout [6]. Healthcare workers experience psychological distress and a "shattering" of
their social identity in the workplace because of the unprecedented emotional-cognitive
demands and challenges to their professional expertise [7]. Likewise, student athletes pre-
vented from participating in team sports and activities experience social identity threat,
reduced well-being, and increased depressive symptoms [8]. In sum, the collective actions
societies are taking to combat COVID-19 are disrupting social roles in ways that lead to social
and psychological disturbance among the citizenry.

The present research examines how the external shock of COVID-19 to social roles is
impacting individual-level *self-authenticity*, the sense that one's thoughts, feelings and actions
are "true to the self" [9–12]. Self-authenticity is an aspect of mental health not to be overlooked
in the best of times, yet is particularly important under COVID-19. Self-authenticity is consid-
ered the foundation of mental health, described as "not simply an aspect or precursor to well-
being but rather the very essence of wellbeing and healthy functioning" ([12], p. 386). For this
reason, safeguarding authenticity implies safeguarding mental health more broadly. Indeed,
*In*authenticity is associated with (but distinct from) reduced self-esteem and increased stress,
anxiety, and depression [10, 12], relationship dissatisfaction [13], and unethical behavior [14].
Safeguarding authenticity is important also because *state* inauthenticity can develop into
*chronic* inauthenticity over time [12]. Thus, COVID-19-related role changes may have lasting
effects on the self, long after the pandemic is over, if otherwise temporary feelings of inauthen-
ticity become one's new normal.

Moreover, a self-authenticity perspective offers a novel path for devising coping strategies
to improve psychological wellbeing under COVID-19. Research finds that people feel more
authentic when they experience self-continuity, stability of the self across time [15]. Drawing
on this research, we propose that COVID-19-related role changes cause *in*authenticity by
*reducing* continuity of the self across time, and we rely on this proposition to test strategies for
coping with COVID-19 that vary in temporal perspective (i.e., past, present, or future). The
results contribute to research on coping with COVID-19 and thus are substantively important
and timely. Indeed, there is nascent research associating emotional wellbeing during COVID-
19 with temporal perspectives. Bodecka et al. [16] show that a chronic tendency to "seize the
moment" reduces negative emotions in women, while Dennis, Ogden and Hepper [17] find
that listing gratitude (which is presumed to elicit a present focus) enhances positive affective
states. That literature raises the possibility that temporal perspective influences wellbeing

during COVID-19, but it does not pit temporal perspectives against each other, nor does it address coping with threats to self-authenticity. The results of the present investigation also contribute to the literature on self-authenticity. Social role disruption is identified as an antecedent to self-inauthenticity. Also, we reveal that previously established mechanisms for coping with inauthenticity (i.e., nostalgia, reflecting on the past; [15]) are less effective during this ongoing pandemic.

## Social role disruption and self-authenticity

Social roles are "positions in social structure that carry social expectations for the behavior of individuals who occupy them" ([18] p. 309; c.f. [19, 20]). These include, for example, the roles of child, parent, friend, romantic partner, or employee. Such roles can be part of people's sense of self and, accordingly, can affect how people relate to and feel about themselves [21]. In particular, social roles can influence how authentic people feel. If a person performs a social role in a manner that is not true to the self (e.g., by subordinating their personal needs in a romantic relationship) they will feel inauthentic [9, 13]. Likewise, people feel inauthentic if they perceive little integration across social roles [11, 22] or adopt many different social roles [23]. Building on this literature, we posit that inauthenticity also is elicited by social role disruptions, like those associated with the COVID-19 pandemic. We conjecture that such disruptions undermine authenticity by reducing self-continuity.

Self-continuity is the subjective perception that one's past, present, and future selves are interconnected [24, 25]. People whose autobiographical narrative is stable over time will subjectively experience self-continuity and feel authentically themselves [15]. In contrast, people whose autobiography lacks temporal continuity will feel inauthentic [15]. Prior research finds that self-continuity is destabilized by life-role transitions. Specifically, the upheaval of a major life role induces liminality, a state during which personal identities are suspended, self-concept becomes ambiguous [26, 27], and self-continuity is reduced [28]. Consider, then, how the magnitude and scope of social role disruptions under COVID-19 might affect self-continuity. As noted, the responsibilities, routines, and contexts relevant to social roles are transforming, with some social roles suspended entirely. Consequently, COVID-19 throws into disarray the temporal continuity underlying social roles; its profound disruptions to the rhythm of daily life render time almost "meaningless", whereby the "past normal" no longer exists, but the "new normal" is as of yet unreached [29]. Accordingly, social role disruptions should heighten inauthenticity because the ways that people enact these roles lack continuity from what they perceived as authentic to these roles before COVID-19.

It is important to note that social role "disruptions" under COVID-19 are not necessarily negative. People may perceive role changes (e.g., homeschooling children) either negatively (e.g., losing childcare) or positively (e.g., spending more time with family). Prior research has linked inauthenticity to both negative actions (e.g., cheating; [14]) and positive actions (e.g., using luxury brands; [30]). We thus posit that the valence people ascribe to role change under COVID-19 and, more generally, to COVID-19 itself, are distinct conceptually and empirically from the extent of role change under COVID-19 and from the effect of role change on inauthenticity. We predict that people feel inauthentic to the extent COVID-19 has changed their social roles, and this occurs independent of (i.e., statistically controlling for) how positively or negatively people feel about COVID-19 (Study 2) or about the role change itself (Studies 3–4).

***Hypothesis 1***: *Social role disruptions under COVID-19 elicit self-inauthenticity.*

We further posit that not all role changes contribute equally to this effect. Rather, changes to a social role should affect inauthenticity to the extent that particular role is central to one's sense of self. Role centrality refers to a role's cognitive prominence and subjective importance

to one's identity [31, 32]. For example, one might perceive their role in the workplace as "merely a job" (i.e., low centrality) or as a reflection of "who I am" (i.e., high centrality). We expect that inauthenticity results primarily from disruptions to roles of the latter type, high-centrality roles. Meanwhile, a social role can undergo great change but exert little effect on inauthenticity if that role is low centrality. So, for example, people for whom the role of employee is high (low) centrality should (should not) feel out of touch with themselves when subjected to, say, a work-from-home protocol.

*Hypothesis 2*: *Disruption to a given social role under COVID-19 elicits self-inauthenticity to the extent that role is central to the self.*

Support for this hypothesis, H2, would highlight that role centrality is an important boundary condition for H1. Support for H2 also would empirically support our conceptual claim that the "self" underlies the effect predicted in H1. That is, establishing that a role's centrality to the self moderates the effect of role disruption on inauthenticity would suggest indeed that it is the link between social roles and the self [1] that enables role disruptions to affect inauthenticity.

## Coping with inauthenticity under COVID-19: The role of temporal perspectives

In sum, the present research tests the arguments that COVID-19-related role changes will cause a person to feel inauthentic to the extent a person experiences disruptions across social roles (H1), and to the extent a person experiences disruptions to a particular role that is central to the self (H2). How, then, should people cope with such threats to authenticity under COVID-19? Individuals have little control over COVID-19's impact on their social roles, but they can control the perspectives they adopt to cope under COVID-19. Here, we are interested specifically in temporal perspectives: the degree to which people think about the past, present, and/or future [33, 34].

Temporal perspective is defined by Zimbardo and Boyd (p.1271) as the "nonconscious process whereby the continual flows of personal and social experiences are assigned to temporal categories, or time frames, that help to give order, coherence, and meaning to those events" [33]. While the current literature identifies up to nine distinct dimensions of temporal perspective [35], these dimensions can be grouped into three main categories: past-focused, present-focused, or future-focused [33, 34]. Through learning, individuals tend to develop a chronic tendency to adopt one temporal perspective more than the others [35]. However, individuals may also learn to adaptively switch temporal focus based on the situation [35, 36], and situational factors themselves may temporarily activate a given temporal focus [34, 37].

Temporal perspectives can affect people's ability to cope in crises, as temporal perspectives can influence people's health, happiness, and sense of self [16, 38–40]. Temporal perspectives thus may affect people's ability to maintain authenticity in the face of social role disruption. To address this possibility, we compare the three main categories of temporal perspective: past-focused, present-focused, and future-focused, which here correspond to focusing on life before COVID-19 began, during COVID-19 (i.e., the present and immediate future), or after COVID-19 ends, respectively.

Past research suggests that people can cope with threats to self-authenticity by reaffirming self-continuity, which is to reaffirm the sense that their past, present, and future selves are interconnected [15]. For this reason, interventions designed to increase self-authenticity often are designed to increase self-continuity, most notably by linking past and present selves [41]. Indeed, absent a direct intervention, people sometimes cope with inauthenticity by seeking out experiences that bolster self-continuity [15]. For example, people who feel inauthentic may

gravitate to products with nostalgic value (e.g., retro products), which makes people feel continuity to their past selves and thereby restores authenticity [15].

However, previously established mechanisms for coping with inauthenticity may not facilitate coping under COVID-19. This is because self-continuity requires reflection on one's past, present, and/or future selves, which involves remembering the past and using it to understand the present and predict the future [42]. But consider this psychological process in light of the COVID-19 pandemic. COVID-19 is ongoing and people cannot confidently expect a return to normalcy. Continuity between the past and present may be forever disrupted. Moreover, with so much uncertainty shrouding the future, people cannot foresee possible future shocks to social systems and roles, and are thus unable to set concrete goals and plans; this renders continuity between the present and future difficult to predict. Focusing on the past and/or the future thus may make salient *discontinuity*, not continuity, of the self over time. Accordingly, rather than advocating to enhance perceptions of self-continuity, we advocate to reduce the salience of self-*dis*continuity during this ongoing pandemic.

One way to reduce the salience of self-discontinuity is to encourage people to focus on the here-and-now, rather than the past or future. Focusing on the present directs attention away from (role) changes over time and, as a result, should reduce the temporal discontinuity of the self—including discontinuity between roles before, during, and after COVID-19. We suggest, therefore, that people can maintain authenticity under COVID-19 by adopting a present-focused temporal perspective.

**Hypothesis 3**: *The effect of COVID-19 related role changes on inauthenticity is attenuated by focusing on the present (but not the past or the future).*

## Study overview

Four studies examine (i) whether social role disruptions related to COVID-19 undermine authenticity and (ii) whether a present-focused coping strategy can attenuate this effect, thereby restoring authenticity. Across studies, COVID-19-related role changes increase *in*authenticity, both when role change is measured (Studies 1, 3 and 4) and when its salience is manipulated (Study 2). Study 2 identifies a moderator of this effect, role centrality. Study 3 then manipulates temporal focus and finds that COVID-19-related role changes have the least impact on inauthenticity when people are present-focused (vs. past- or future-focused). These findings suggest that prompting people to focus on the here-and-now is an effective way to cope with COVID-19-related role changes. Study 4 measures people's baseline levels of past-, present- and future-focused coping during the pandemic, and tests these individual differences as moderators. We find that people who tend to maintain a present-focused (as opposed to a past- or future-focused) temporal perspective under COVID-19 are best able to maintain authenticity. Notably, these effects occur independent of how positively/negatively people feel about COVID-19 (Study 2), or how positively/negatively people feel about the social role disruptions (Studies 3 and 4).

## Sample sizes, statistical analyses & procedure disclosure

For Studies 1–3 (conducted online), power analyses were conducted to determine the minimum sample size necessary to achieve moderate power ($1$-$\beta$ = .80) and medium effect sizes ($f^2$ = .15) for two-tailed analyses. We recruited a larger sample size than the minimum requirement and ensured that each condition had $n > 50$ per condition. For Study 4, participants were recruited based on having completed a prior, unrelated study in which we measured chronic authenticity (among those who had completed a prior study, we aimed to recruit as many as possible to participate). Post hoc power analysis revealed that the resulting sample

size was sufficient for the study design. A sensitivity power analysis based on the actual sample size was conducted and reported in each study. Our sample sizes of all studies were sufficient to detect small to medium effects. Studies 1–3 recruited convenience samples of American residents from Amazon's Mechanical Turk (MTurk). Study 4 recruited participants (students and staff) from a university in Hong Kong. For all studies, data collection was completed prior to conducting any data analyses, and no additional data were collected after analyses began.

We conducted all statistical analyses using SPSS Version 26. For all linear regression analyses, we confirmed that the residuals were distributed normally. The Normal P-P Plot and Scatterplot of Residuals for the key regressions are reported in S1 Appendix in S1 File. We also tested for multi-collinearity in all regressions and reported the Variance Inflation Factor (VIF) in the result sections of the studies. All VIF were smaller than 10, indicating that multi-collinearity was not an issue. Across studies, we reported all manipulations and measures pertinent to the study (see stimuli in S2 Appendix in S1 File), and all participant exclusions. Additional analyses (e.g., analyses with different operationalizations of measures) and exploratory analyses (e.g., individual differences measured as potential covariates and moderators) are discussed in S3 to S5 Appendices. Datasets for all studies can be found at https://osf.io/f6abv/?view_only=5bf2b36dc9d448f2a1638ab698c9a28f.

## Study 1: Role changes and self-inauthenticity

Study 1 tests whether social role disruptions under COVID-19 are positively associated with feelings of inauthenticity (H1). Because inauthenticity tends to correlate with low self-esteem [12], we test this hypothesized relationship while controlling for self-esteem.

### Method

This research has been approved by the Institutional Review Board of Hong Kong University of Science and Technology (GRF-16501420) and the Research Ethics Committee at City, University of London (ETH2021-0291). In this study and the following studies, we obtained informed consent from all participants at the outset of the studies.

MTurkers ($N = 224$) completed a survey in exchange for monetary payment. Participants viewed a list of nine roles (parent, child, employee, student, spouse/partner, friend, sibling, other) and rated the extent to which each role that applied to them had changed since the COVID-19 outbreak began (1 = *my role has no change*, 7 = *my role has significant changes*). Participants selected "N/A" for inapplicable roles (e.g., participants select "N/A" for "parent" if they are not a parent). Participants then completed an Authenticity Scale [12], a measure for chronic authenticity personality. The scale includes three subscales: self-alienation, inauthentic behaviors, and susceptibility to external influences. We administered the entire scale in all studies but used the 4-item self-alienation subscale as the dependent variable (DV). This is because self-alienation describes "how inauthenticity might feel to the person experiencing it" ([10], p. 277), and it has been used to measure state inauthenticity in prior research (e.g., [14] [43]). We reported additional analyses using the full scale as the DV in S3 Appendix in S1 File.

Participants also completed an attention check embedded in the inauthenticity scale, a 10-item self-esteem scale measure ([44]; 7-point scale, $\alpha = .88$, $M = 4.79$, $SD = 1.26$), demographics (e.g., age, gender, income), and COVID-19-related questions (e.g., whether they know someone who has COVID-19; whether they are essential workers) as potential covariates. Excluding nine who failed the attention check and three who selected "N/A" for all roles left $N = 212$ for analyses. We summarized the demographic characteristics of participants across studies in S1 Table in S1 File.

## Results and discussion

**Hypothesis testing.**　We computed a role change index ("role change" hereafter) by averaging the ratings of all roles applicable to a participant (7-point scale, $\alpha$ = .92, $M$ = 4.16, $SD$ = 1.90). The index measured the overall levels of role change one experienced during COVID-19. Descriptive summary of the role change measure is reported in S2 Table in S1 File. We computed an inauthenticity index ("inauthenticity" hereafter) by averaging the ratings of the four items in the self-alienation subscale ([12]; 7-point scale, $\alpha$ = .94, $M$ = 3.74, $SD$ = 2.01).

To test whether role change positively predicted inauthenticity (H1), we calculated a Pearson's correlation between the role change index and inauthenticity index. As predicted, we found a significant positive correlation ($r$ = .59, $p < .001$, 95% CI = [.493, .671]). We also separately tested the correlations for each role. While effect sizes varied (perhaps because the individual roles varied in role centrality; see Study 2), role change and inauthenticity correlated positively for each role (results reported in S3 Table in S1 File). Thus, whether we look across roles or within a single role, the greater the magnitude of role change, the more inauthentic people feel.

Furthermore, we tested whether role change affects inauthenticity controlling for self-esteem. We conducted a linear multiple regression on inauthenticity using the role change index and self-esteem as independent variables (Eq 1). A sensitivity power analysis based on our sample size (assuming $\alpha$ = .05, two tailed, power = 80%, multiple regression with two predictors) revealed $f^2$ = .04 as the required effect size, indicating that our sample size was sufficient to detect a small to medium effect. The regression fulfilled normality assumptions of linear regressions (see S1 Appendix in S1 File for residual plots). Consistent with H1, results of the regression analysis (*adjusted $R^2$* = .63, $f^2$ = 1.703; $F(2, 209)$ = 180.69, $p < .001$) revealed that role change positively predicted inauthenticity ($\beta$ = .36, $t(209)$ = 7.98, $p < .001$, 95% CI = [.549, .910], VIF = 1.18) controlling for self-esteem. In addition to corroborating H1, this result establishes discriminant construct validity between inauthenticity and self-esteem. Consistent with prior research [12], self-esteem negatively predicted inauthenticity ($\beta$ = -.58, $t(209)$ = -12.80, $p < .001$, 95% CI = [-1.350, -.990], VIF = 1.18).

$$\text{inauthenticity} = \alpha_i + \beta_1 \text{ role change} + \beta_2 \text{ self–esteem} + \varepsilon_i \qquad \text{Eq 1}$$

**Robustness testing.**　Given the correlational design of this study, we tested robustness of our effects by controlling for demographic variables and other COVID-19 related factors as covariates. The positive effect of role change on inauthenticity held (see S4 Table in S1 File), providing further confidence that social role change due to COVID-19 is linked to inauthenticity, per H1.

## Study 2: Role centrality as a moderator

The key objective of Study 2 is to test the moderating effect of role centrality (H2). This moderator is important because it captures the importance of a given social role to the self and, for this reason, allows us to test a key assumption of our conceptual process model: namely, that COVID-19-related role changes influence inauthenticity because (and therefore, when) social roles are integral to one's sense of self [1]. Accordingly, a role's centrality to the self should moderate our effect. Moreover, examining role centrality provides nuance to our empirical findings. The other studies reported herein establish that, across all social roles, the greater the overall change, the more inauthentic people feel. Study 2 endeavors to show that not all roles contribute equally to this effect. Changes in low centrality roles should not affect self-inauthenticity, implying that role centrality is a boundary condition.

Study 2 also adopts a different study design. In Study 1, we showed a positive relationship between COVID-19-related role change and inauthenticity, per H1, which held controlling for self-esteem, demographics, and other factors. In the current study, we further test the effect of COVID-19-related role change on inauthenticity in a more controlled design, by experimentally manipulating, rather than measuring, the key independent variable, role change. It is not feasible to experimentally manipulate the degree to which COVID-19 disrupts a social role, but we can experimentally manipulate the *salience* of social role disruption. Accordingly, Study 2 draws participants' attention to a social role that has significantly changed (vs. remained constant) under COVID-19. Per H2, we predict that making salient role change (vs. constancy) will increase inauthenticity if the role is high-centrality, and this effect will attenuate if the role is low-centrality.

## Method

MTurkers ($N$ = 304) completed a 2 (role: changed vs. constant) by role centrality (continuous measure) between-subjects experiment in exchange for monetary payment. We manipulated the salience of role change due to COVID-19. From nine given roles, participants selected the roles that changed most and least under COVID-19. Those assigned to the changed (vs. constant) role condition described how their most (vs. least) changed role had changed (vs. remained constant). Then, as our dependent measure, participants completed an Authenticity Scale ([12]; as before, the self-alienation subscale was used as the DV: 7-point, $\alpha$ = .95, $M$ = 3.85, $SD$ = 2.07), an attention check embedded therein, a self-esteem scale ([44]; 7-point, $\alpha$ = .88, $M$ = 4.80, $SD$ = 1.29), a single-item mood measure (*How do you feel right now?*; 1 = *very negative*, 7 = *very positive*; $M$ = 5.65, $SD$ = 1.21), a manipulation check (*To what extent do you feel that the different roles you play in life are impacted by COVID-19?*), and demographics.

To test the moderating effect of role centrality, participants then completed Cameron's Strength of Identification (SOI) scale, which includes a role centrality subscale ([32]; 7-point, $\alpha$ = .66, $M$ = 4.20, $SD$ = 1.32). The SOI scale includes three subscales: centrality (i.e., subjective importance of the social identity), affect (i.e., feeling about the social identity), and tie (i.e., psychological ties with group members). The latter two subscales capture interpersonal aspects of group membership and thus are not as theoretically relevant here, but the key results do replicate using the full scale instead of the centrality subscale as the moderator (S4 Appendix in S1 File). Participants then completed other individual difference measures that are not relevant to our predictions but included in the data files (https://osf.io/f6abv/?view_only=5bf2b36dc9d4 48f2a1638ab698c9a28f; analyses available upon request). Next, we asked participants to rate how positively/negatively they felt about COVID-19's effect on the role they wrote about. We measured this because our manipulation might affect feelings toward COVID-19's impact (e.g., people who write about a changed vs. constant role might feel more negatively). Excluding those who failed the attention check ($n$ = 15; the number of exclusions did not differ across the two role change conditions: $X^2$(1, $N$ = 304) = .717, $p$ = .397), left $N$ = 289 for analyses.

## Results

**Preliminary analyses.** We conducted three analyses before proceeding to test H2. First, a one-way ANOVA confirmed that the changed (vs. constant) role condition perceived greater life changes under COVID-19 ($M_{change}$ = 5.29, $SD$ = 1.36, vs. $M_{constant}$ = 4.98, $SD$ = 1.63; $F$(1, 287) = 3.13, $p$ = .078, 95% CI = [-.035, .659], $d$ = .21). Thus, the role change manipulation was successful, albeit at marginal statistical significance. For descriptive purposes, S2 Table in S1 File lists each of the nine roles we asked about and the corresponding percentage of participants who indicated that the role was the "most changed" and the "least changed" by COVID-19.

A second one-way ANOVA showed that the changed (vs. constant) role condition felt more negatively about the impact of COVID-19 ($M_{change}$ = 4.57, $SD$ = 1.69, vs. $M_{constant}$ 5.23, $SD$ = 1.34; $F(1, 287)$ = 13.35, $p < .001$, 95% CI = [-1.015, -.304], $d$ = .43). We thus kept this variable as a covariate in the regression analysis that tests H2.

A third one-way ANOVA confirmed that the individual difference measure, role centrality, was not affected by our experimental manipulation of role change ($M_{changed}$ = 4.13 $SD$ = 1.38, vs. $M_{constant}$ = 4.27, $SD$ = 1.26; $p$ = .34, 95% CI = [-.454, .158], $d$ = -.11). We thus tested proceeded to test H2 using role centrality as a moderator.

**Hypothesis testing.**   To test whether role centrality moderates the effect of social role change (vs. constancy) on inauthenticity (H2), we conducted a linear multiple regression on inauthenticity, using the role manipulation (1 = changed, -1 = constant), role centrality (standardized), and their interaction as independent variables, and valence of COVID-19's impact (standardized) as a covariate (Eq 2). A sensitivity power analysis based on our sample size (assuming $\alpha$ = .05, two tailed, power = 80%; multiple regression with four predictors) revealed $f^2$ = .03 as the required effect size, indicating that our sample size was sufficient to detect a small to medium effect. The regression fulfilled normality assumptions of linear regressions (S1 Appendix in S1 File).

In support of H2, results of the analysis (*Adjusted $R^2$* = .17, $f^2$ = .199; $F(4, 284)$ = 15.30, $p < .001$) revealed the predicted 2-way interaction between the manipulated role change variable and the measured role centrality variable on feelings of inauthenticity ($\beta$ = .12, $t(284)$ = 2.20, $p$ = .028, 95% CI = [.027, .470], VIF = 1.02). Results also showed main effects of role centrality ($\beta$ = -.26, $t(284)$ = -4.77, $p < .001$, 95% CI = [-.762, -.317], VIF = 1.03) and valence of COVID-19's impact ($\beta$ = .36, $t(284)$ = 6.45, $p < .001$, 95% CI = [.514, .966], VIF = 1.06).

$$\text{inauthenticity} = \alpha_i + \beta_1 \text{ role conditions} + \beta_2 \text{ role centrality} + \beta_3 \text{ role change conditions}$$
$$* \text{ role centrality} + \beta_4 \text{ valence of COVID–19's impact} + \varepsilon_i \qquad \text{Eq 2}$$

To decompose the 2-way interaction, we conducted a spotlight analysis using SPSS PROCESS Model 1. As theorized, role change (vs. constancy) increased inauthenticity for high-centrality roles (+1 $SD$ on role centrality: $b$ = .33, $SE$ = .16, $t(284)$ = 2.05, $p$ = .041, 95% CI = [.014, .646]), but not low-centrality roles (-1 $SD$ on role centrality: $b$ = -.17, $SE$ = .16, $t(284)$ = 1.04, $p$ = .297, 95% CI = [-.482, .148]; Fig 1). Per H2, these results suggest that role centrality represents a boundary condition for the effect of COVID-19-related role changes on inauthenticity.

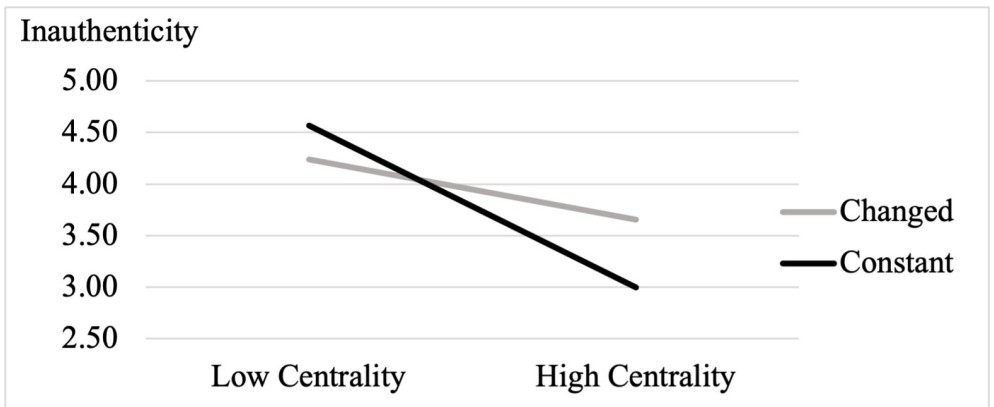

**Fig 1. Interaction of role change and role centrality on state inauthenticity in Study 1.**

Disruptions to high-centrality roles have a significant impact on inauthenticity but low-centrality roles can undergo much change and have little impact on inauthenticity.

**Discriminant validity and robustness testing.** To test discriminant validity among inauthenticity, self-esteem, and mood, we conducted additional linear multiple regressions on self-esteem and mood, respectively, using the same predictors as in Eq 2. We did not observe the same role change X role centrality interaction on these DVs, as we did on inauthenticity. These results suggested that making salient a central, changed (vs. constant) social role makes people feel inauthentic; but it does not affect their self-esteem or mood.

Moreover, we conducted two additional regression analyses on inauthenticity, while keeping self-esteem and mood as a respective covariate. The predicted role change X role centrality interaction on inauthenticity held, attesting to the robustness of our effects (S5 Table in S1 File).

## Discussion

Supporting our prediction, manipulating the salience of role change (vs. constancy) related to COVID-19 increases feelings of inauthenticity, especially when the changed role is central to the self. The moderating effect of role centrality speaks to the process of our effects. That is, COVID-19-related role changes are disruptive to self-authenticity when (and therefore because) the social roles are central to the self. Moreover, the results of this study suggest that not all changes in life elicit feelings of inauthenticity to the same extent. While the overall levels of changes across social roles are linked to inauthenticity (per Study 1), the changes in a specific role alone may not affect self-authenticity if the role is not central to the self (per Study 2).

Studies 1 and 2 established the effect of role changes on inauthenticity. In Studies 3 and 4, we test temporal perspectives as coping strategies.

## Study 3: Temporal perspective as a coping strategy

The key object of Study 3 is to experimentally test coping strategies that can potentially alleviate the effects of COVID-19 on feelings of inauthenticity. Rather than testing specific coping strategies tailored to specific social roles, we instead sought to test general coping strategies that are broadly applicable, in that their effectiveness should not depend on the specific role that has changed. Moreover, as mentioned, we sought to test coping strategies that are relevant to self-authenticity *per se* and, in particular, to the temporal component underlying authenticity. To this end, we manipulate and compare past-, present-, and future-focused coping.

While manipulating the salience of a specific changed role allowed us to test process by moderation (as in Study 2), the reality of people's lives under COVID-19 is that they must adapt to changes in multiple roles. Thus, a more ecologically valid measure of role change is one that captures the magnitude of change across all of a given individual's social roles (as in Study 1). As a result, rather than making salient a specific changed role (as in Study 2), we again measure the overall changes across a given individual's social roles and test the effect of temporal focus in coping with these changes. We predict that the effect of overall role change on feelings of inauthenticity will be attenuated when individuals engage in present-focused coping compared to either past-focused or future-focused coping.

## Method

MTurkers (*N* = 452) completed a pre-registered (pre-registration form can be found: https://osf.io/yavr3?view_only=508b5fb3cae34bd496558ae7671affe5), 3 (temporal focus: past, present, future) by role change (continuous measure) between-subjects study. Participants first completed the role change measure. As in Study 1, they viewed a list of nine roles (parent, child,

employee, student, spouse/partner, friend, sibling, other) and rated the extent to which each role that applied to them had changed due to the COVID-19 outbreak (1 = *my role has no change*, 7 = *my role has significant changes*). Participants selected "N/A" for inapplicable roles. A role change index was created by averaging the rating across all roles applicable ($\alpha$ = .94, *M* = 3.83, *SD* = 1.81).

Then, participants were randomly assigned to one of three temporal focus conditions. The present- (vs. past-; vs. future-) focus condition wrote about how COVID-19 makes them focus on *the present, the one day at a time* (vs. *the past, life before COVID-19*; vs. *the future, life after COVID-19*). As the dependent variable, participants then completed the Authenticity Scale ([12]; as before, the self-alienation subscale was used as DV: $\alpha$ = .95, *M* = 3.11, *SD* = 1.95), an attention check embedded therein, self-esteem ([44]; $\alpha$ = .89, *M* = 5.07, *SD* = 1.29), mood (as in Study 2; *M* = 5.40, *SD* = 1.33), demographics, and valence of role change (i.e., *how positive or negative were the changes in their roles due to COVID-19; 1 = very negative, 7 = very positive; M* = 4.49, *SD* = 1.45). Because behaviors that elicit inauthenticity need not be negative in nature (e.g., [30]), we expect that role change affects inauthenticity independently of the effect of the valence of role change. Accordingly, we also do not expect the coping effect of a present-focus temporal perspective to be driven by the perceived valence of role change. To rule out these possibilities, we measured valence of role change to keep as a covariate in our analyses.

Finally, participants reported difficulty of the writing task (no difference across conditions, all contrasts *p*s > .36) and individual differences (for exploratory analyses, e.g., mindfulness, avoidant coping; S5 Appendix in S1 File). Excluding four who selected "N/A" for all roles and 13 who failed the attention check left N = 435 for analyses (the number exclusions did not differ across conditions: $X^2$(2, N = 452) = .851, *p* = .653).

## Results

**Preliminary analyses.** An ANOVA confirmed that role change did not differ by temporal focus conditions (*F*(2, 432) = .80, *p* = .45; contrast comparisons: $M_{present}$ = 3.99, *SD* = 1.76 vs. $M_{past}$ = 3.75, *SD* = 1.82; *p* = .249, 95% CI = [-.175, .671], *d* = .14; $M_{present}$ = 3.99, *SD* = 1.76 vs. $M_{future}$ = 3.77, *SD* = 1.86; *p* = .297, 95% CI = [-.198, .647], *d* = .12; past vs. future, *p* = .909, 95% CI = [-.436, .389], *d* = .01). We thus proceeded to test H3, the interaction of temporal focus and role change, using this measure of role change as an independent variable.

**Hypothesis testing.** To test the hypothesis that present focus attenuates the effect of role change on inauthenticity (H3), we conducted a linear multiple regression. Treating present-focus condition as the benchmark, we regressed inauthenticity on role change (standardized), past-focus (1 = yes, 0 = otherwise), future-focus (1 = yes, 0 = otherwise), role change X past-focus interaction, and role change X future-focus interaction, with valence of change as a covariate (Eq 3). A sensitivity power analysis based on our sample size (assuming $\alpha$ = .05, two tailed, power = 80%, six predictors) revealed $f^2$ = .02 as the required effect size, indicating that our sample size was sufficient to detect a small effect. The regression fulfilled normality assumptions (S1 Appendix in S1 File).

In support of H3, results of the analysis (*Adjusted $R^2$* = .39, $f^2$ = .65; *F*(6, 428) = 46.32, *p* < .001) revealed both a role change X past-focus interaction ($\beta$ = .14, *t*(428) = 2.53, *p* = .01, 95% CI = [.105, .834], VIF = 2.20) and a role change X future-focus interaction ($\beta$ = .15, *t*(428) = 2.64, *p* = .01, 95% CI = [.124, .845], VIF = 2.25). We also observed main effects of role change ($\beta$ = .38, *t*(428) = 5.29, *p* < .001, 95% CI = [.461, 1.006], VIF = 3.57), valence of role change ($\beta$ = .18, *t*(428) = 4.62, *p* < .001, 95% CI = [.202, .500], VIF = 1.07), and past-focus ($\beta$ = .09, *t*(428)

= 1.97, $p$ = .050, 95% CI = [.000, .716], VIF = 1.39); Fig 2).

$$\text{inauthenticity} = \alpha_i + \beta_1 \text{ role change} + \beta_2 \text{ past focus} + \beta_3 \text{ future focus} + \beta_4 \text{ role change}$$
$$* \text{ past focus} + \beta_5 \text{ role change} * \text{future focus} + \beta_6 \text{ valence of the role change}$$
$$+ \varepsilon_i \qquad \text{Eq 3}$$

To decompose the interactions, we conducted both slope and spotlight analyses using SPSS PROCESS Model 1. The slope analyses revealed that role change increased inauthenticity in all three conditions, per H1. But, per H3, the slope of the effect of role change on inauthenticity was weakest in the present-focus condition (present: $b$ = .73, $SE$ = .14, $t$(428) = 5.29, $p < .001$, 95% CI = [.461, 1.006]; past: $b$ = 1.20, $SE$ = .13, $t$(428) = 9.56, $p < .001$, 95% CI = [.956, 1.451]; future: $b$ = 1.22, $SE$ = .12, $t$(428) = 9.91, $p < .001$, 95% CI = [.977, 1.460]). These results suggest that people who focus on the present are best able to safeguard their feelings of authenticity across varying levels of COVID-19-related social role disruptions.

Further corroborating H3, spotlight analyses suggested that when experiencing a high level of role change (+1 $SD$), the present-focus condition felt less inauthentic compared to the past- ($b$ = -.83, $SE$ = .26, $t$(428) = -3.23, $p$ = .001, 95% CI = [-1.332, -.323]), and the future- ($b$ = -.44, $SE$ = .25, $t$(428) = -1.72, $p$ = .09, 95% CI = [-.937, .063]) focus conditions. These effects were not observed when the level of role change was low (-1 $SD$: present vs. past: $b$ = .11, $SE$ = .26, $t$(428) = .43, $p$ = .670, 95% CI = [-.405, .629]; present vs. future: $b$ = .53, $SE$ = .26, $t$(428) = 2.03, $p$ = .043, 95% CI = [.017, 1.048]). These results suggest that, among people facing the greatest upheaval to their social roles under COVID-19, those who focus on the past or future feel significantly more inauthentic than those who focus on the present.

**Discriminant validity and robustness testing.** To test discriminant validity among inauthenticity, self-esteem, and mood, as in Study 2, we conducted additional linear multiple regressions on self-esteem and mood, respectively, using the same predictors in Eq 3. We did not observe significant interactions on mood but observed the interactions on self-esteem. To further test discriminant validity and to test robustness of our effects, we conducted additional regression analyses on inauthenticity, while keeping self-esteem and mood, respectively, as an additional covariate, as well as testing the effects without any covariate. The observed role change X past focus and role change X future focus interactions (i.e., present focus was the benchmark) held across models, though the role change X past focus interaction became marginal when controlling for self-esteem (S6 Table in S1 File).

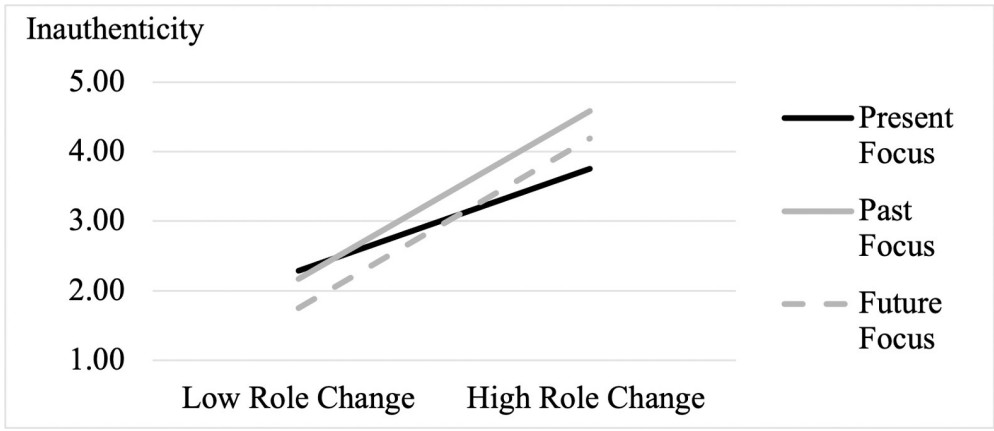

**Fig 2. Interaction of role change and present-focus on state inauthenticity in Study 2.**

**Auxiliary analyses of valence of role change as a second moderator.** In the above analyses, we controlled for valence of role change as a covariate. Results of those analyses suggested that the effect of role change on inauthenticity, as well as the coping effect of present-focused orientation, were *not* driven by the perceived negativity or positivity of the changes. These results suggest that the benefit of present-focus coping does not depend on the emotional valence (negative to positive) people experience toward the role change. this argument also can be examined by testing the three-way interactions of (a) role change X past focus X valence of change, and (b) role change X future focus X valence of change (i.e., keeping present focus as the benchmark). The results yielded no three-way interactions, but the two 2-way interactions of role change X past focus and role change X future focus still held (detailed results see S7 Table in S1 File). Thus, as theorized earlier, and consistent with prior research [14, 30], the benefit of maintaining a present focus (i.e., present focus mitigated the effect of role change on inauthenticity), occurred regardless of whether the role change itself was perceived positively (e.g., homeschooling is an opportunity to spend quality time with children) or negatively (e.g., homeschooling is a burdensome loss of childcare).

## Discussion

Study 3's experimental design pits three temporal coping strategies against one another and allows us to make causal inferences about the efficacy of these strategies. The results show that focusing on the present, as opposed to the past or the future, can mitigate the feelings of inauthenticity otherwise elicited by social role disruptions under COVID-19, per H3.

These results imply that focusing on the here-and-now protects against threats to authenticity. Following up on this result, an interesting question is whether people do this naturally. To address this, Study 4 examines the extent to which people naturally have adopted each of the three temporal perspectives during COVID-19. We expect that the effect of COVID-19-related role changes on inauthenticity can be offset among people who naturally stay present-focused. Whether or not people sense the benefits of staying present focused as a coping mechanism is an empirical question, which we address by examining whether present-focus is people's dominant temporal focus under COVID-19.

## Study 4: Individual difference in temporal focus

Study 4's first objective is to replicate the effect of role change on inauthenticity (H1) using a different population: Hong Kong residents. Hong Kongers differ from Americans by culture (e.g., self-construal, independence/interdependence), by the onset of COVID-19 and government responses (earlier in HK than USA), and by the types of roles affected (Hong Kong sample draws from the university community whereas USA sample draws from the general public). Yet we predict that H1 should hold. Moreover, we measure and test this effect controlling for pre-COVID-19 inauthenticity levels.

Second, Study 4 tests whether the effect predicted in H1 is moderated by people's tendency to be present-focused (and not past- or future-focused; H3). Whereas we tested H3 in Study 3 by experimentally manipulating the moderator, temporal perspective, in Study 4 we tested moderation by individual differences in temporal perspective.

## Method

We contacted 996 university staff and students from a major Hong Kong university. These university members had all completed an unrelated survey one year ago (April 2019), in which they reported chronic inauthenticity. We used this pre-COVID-19 chronic inauthenticity as a covariate in our analyses to ensure that any significant association between role change and

inauthenticity observed in our data is not due to chronically inauthentic people changing roles more (i.e., reverse causality). We invited these participants to participate in a follow-up survey for a chance to win a prize, and 299 responded (30% response rate).

In the follow-up survey (administered four months into the outbreak of COVID-19 in Hong Kong), participants completed measures of authenticity ([12]; as before, self-alienation subscale was the DV: $\alpha$ = .82, $M$ = 3.45, $SD$ = 1.20), self-esteem ([44]; 4-point, $\alpha$ = .85, $M$ = 2.77, $SD$ = .44), role change (as in Studies 1 & 3; 7-point, $\alpha$ = .93, $M$ = 3.48, $SD$ = 1.50), and valence of role change (as in Study 3; 7-point single item, $M$ = 4.01, $SD$ = 1.24). To ensure that we capture the natural levels of inauthenticity and self-esteem without making salient COVID-19, we administered the measures of inauthenticity and self-esteem *before* administering the measure of role change.

Next, participants reported how often they thought about life in the i) *past*, before COVID-19 ($M$ = 3.13, $SD$ = .93), ii) *present*, during COVID-19 ($M$ = 3.49, $SD$ = .87), and iii) *future*, after COVID-19 ($M$ = 3.54, $SD$ = .93). This measure of temporal focus was taken amid other individual difference measures not reported here but included in the data files (https://osf.io/f6abv/?view_only = 5bf2b36dc9d448f2a1638ab698c9a28f; analyses available upon request). Among those measures, we also asked about changes in physical appearance due to COVID-19 and the perceived valence of these changes (see S8 Table in S1 File for analyses and results). Finally, participants reported demographics. Excluding one who selected "N/A" for all roles and seven whose participant IDs did not match between surveys left 291 respondents.

## Results

**Hypothesis testing (H1): Main effect.** A Pearson's correlation analysis revealed that role change and inauthenticity were positively correlated ($r$ = .21, $p <$ .001, 95% CI = [.102, .321]). This finding supports H1 and replicates Study 1. Correlations between inauthenticity and change of each role vary in significance (see S9 Table in S1 File). This finding might indicate that some role changes affect inauthenticity more than others, which would correspond with the results of Study 2.

To further test this relationship while controlling for related constructs, we conducted a linear multiple regression on inauthenticity, against role change, self-esteem, valence of change, and pre-COVID-19 inauthenticity (Eq 4). A sensitivity power analysis based on our sample size (assuming $\alpha$ = .05, two tailed, power = 80%, four predictors) revealed $f^2$ = .03 as the required effect size, indicating that our sample size was sufficient to detect a small to medium effect. The regression fulfilled normality assumptions (S1 Appendix in S1 File). Results of the analysis (*Adjusted $R^2$* = .45, $f^2$ = .82; $F(4, 286)$ = 59.17, $p <$ .001) yielded that, role change positively predicted inauthenticity ($\beta$ = .17, $t(286)$ = 3.72, $p <$ .001, 95% CI = [.062, .203], VIF = 1.05), while controlling for self-esteem ($\beta$ = -.43, $t(286)$ = -9.02, $p <$ .001, 95% CI = [-1.428, -.917], VIF = 1.21), valence of the role change ($\beta$ = -.04, $t(286)$ = .923, $p$ = .36, 95% CI = [-.125, .045], VIF = 1.04), and individuals' pre-COVID-19 chronic level of inauthenticity ($\beta$ = .32, $t(286)$ = 6.70, $p <$ .001, 95% CI = [.205, .376], VIF = 1.22). Supporting H1, the results suggested that the effect of role change on inauthenticity was robust and distinct from the effect of self-esteem, valence of the role change, and individuals' chronic level of inauthenticity.

$$\text{inauthenticity} = \alpha_i + \beta_1 \text{ role change} + \beta_2 \text{ self−esteem} + \beta_3 \text{ valence of role change}$$
$$+ \beta_4 \text{ pre−COVID−19 inauthenticity} + \varepsilon_i \qquad \text{Eq 4}$$

**Hypothesis testing (H3): Temporal focus moderator.** Next, to test the moderating effect of individuals' natural levels of present-focused orientation (H3), we conducted a linear

multiple regression on inauthenticity, against role change, present-focus, and their interaction, keeping self-esteem, valence of role change, and pre-COVID-19 inauthenticity as covariates (all scale measures were standardized; Eq 5). A sensitivity power analysis based on our sample size (assuming α = .05, two tailed, power = 80%, six predictors) revealed $f^2$ = .03 as the required effect size, indicating that our sample size was sufficient to detect a small to medium effect. The regression fulfilled normality assumptions (S1 Appendix in S1 File).

Consistent with H3, results of the analysis (*Adjusted $R^2$* = .46, $f^2$ = .85; $F(6, 284)$ = 41.45, $p <$ .001) yielded a marginal role change by present-focus interaction ($\beta$ = -.08, $t(284)$ = -1.71, $p$ = .088, 95% CI = [-.185, .013], VIF = 1.01), and main effects of role change ($\beta$ = .19, $t(284)$ = 4.19, $p <$ .001, 95% CI = [.121, .335], VIF = 1.10), present-focus ($\beta$ = -.10, $t(284)$ = -2.14, $p$ = .03, 95% CI = [-.218, -.009], VIF = 1.05), self-esteem ($\beta$ = -.43, $t(284)$ = -9.06, $p <$ .001, 95% CI = [-.629, -.405], VIF = 1.21), and pre-COVID-19 inauthenticity ($\beta$ = .32, $t(284)$ = 6.72, $p <$ .001, 95% CI = [.272, .498], VIF = 1.22; S10 Table in S1 File).

$$\text{inauthenticity} = \alpha_i + \beta_1 \text{ role change} + \beta_2 \text{ present focus} + \beta_3 \text{ role change} * \text{present focus}$$
$$+ \beta_4 \text{ self} - \text{esteem} + \beta_5 \text{ valence of role change}$$
$$+ \beta_6 \text{ pre}-\text{COVID}-19 \text{ inauthenticity} + \varepsilon_i \qquad \text{Eq 5}$$

To decompose the interaction, we then conducted two simple effect analyses using SPSS PROCESS Model 1. First, we found that role change increased inauthenticity to a lesser extent among those high in present focus (+1 *SD* on present-focus: $b$ = .14, *SE* = .07, $t(284)$ = 1.93, $p$ = .054, 95% CI = [-.003, .283]), compared to those low in present focus (-1 *SD* on present-focus: $b$ = .31, *SE* = .08, $t(284)$ = 4.14, $p <$ .001, 95% CI = [.165, .465]). Consistent with Study 3, this result suggests that, while role change increased inauthenticity overall, the effect is attenuated among those focused on the present.

Second, we found that when experiencing a high level of role change (+1 *SD*), individuals' tendencies to focus on the present reduced inauthenticity ($b$ = -.20, *SE* = .07, $t(284)$ = -2.73, $p$ = .007, 95% CI = [-.344, -.056]). But present focus had no effect on inauthenticity when there was little role change (-1 *SD*: $b$ = -.03, *SE* = .07, $t(284)$ = -.35, $p$ = .73, 95% CI = [-.171, .120]; Fig 3).

Furthermore, we conducted additional regression analyses on inauthenticity, replacing present-focus with past-focus and future-focus, respectively, as the moderator. The results did not yield similar coping effects as observed for present-focus (S10 Table in S1 File).

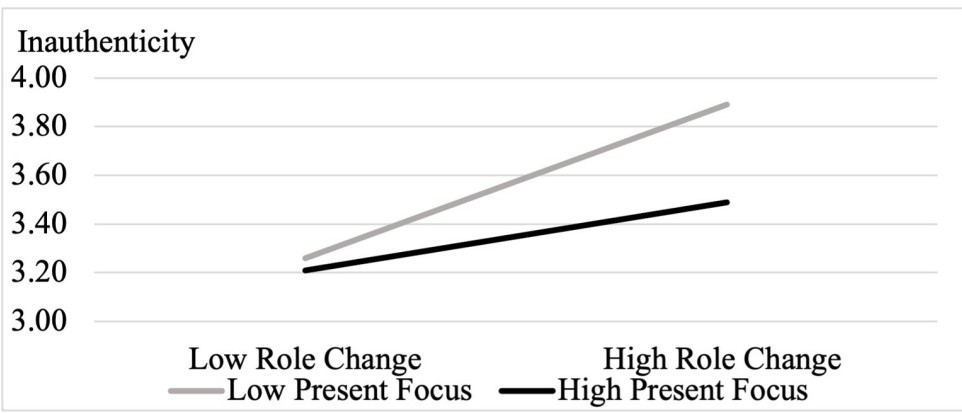

**Fig 3. Interaction of role change and present-focus on inauthenticity in Study 4.**

**Robustness testing & exploratory analysis.**   As in Study 1 (also a correlational survey), we tested robustness of our effects with additional covariates. In a multiple linear regression (S8 Table in S1 File), we included measures related to changes in physical appearance during COVID-19, valence associated with such changes, and demographics, as covariates. Our key predicted effects on inauthenticity held, attesting to their robustness. Moreover, the analysis revealed an interesting empirical point. While changes in *social roles* positively predicted self-inauthenticity, changes in *physical appearance* had no effect. While people's physical appearance generally is important to the sense of self, these findings suggested that during COVID-19, changes in social roles might be more disruptive than changes in physical appearance to people's self-authenticity.

**Baseline temporal perspective.**   To determine whether there was a dominant temporal perspective under COVID-19, we performed paired-samples t-tests on present-focus vs. past-focus and future-focus. We found that present-focus ($M = 3.49$, $SD = .87$) was equally common as future-focus ($M = 3.54$, $SD = .93$, $t(290) = -.88$, $p = .379$, $d = -.04$, 95% CI = [-.178, .068], and both were more common than past-focus ($M = 3.13$, $SD = .93$; present vs. past: $t(290) = 5.33$, $p < .001$, $d = .31$, 95% CI = [.228, .494]; future vs. past: $t(290) = 6.24$, $p < .001$, $d = .35$, 95% CI = [.285, .547]), speaking to the need to prompt present-focus (e.g., Study 3).

## Discussion

Study 4 studies the effect of COVID-19-related role change on inauthenticity (H1) in a different population: a university community in Hong Kong, rather than an online panel in the U.S. A. (as in Studies 1–3). Building on Studies 1–3, Study 4 again supports H1. Study 4 further shows that H1 holds controlling for pre-COVID-19 inauthenticity, speaking against the possibility that the association between role change and inauthenticity is due to chronically inauthentic people experiencing more changes to their roles (i.e., reverse causality). Moreover, Study 4 establishes moderation by temporal perspective, such that people's natural tendency to adopt a present-focus offsets the effect of COVID-19-related role change on inauthenticity. This finding conceptually replicates Study 3 and supports H3.

## General discussion

### Study summary

In four studies, we demonstrate that an external shock (COVID-19) that disrupts social systems and social roles can undermine individual-level inauthenticity. We posit that this effect occurs because social roles are part of the self. Accordingly, Study 2 shows that the more central (important) the role to one's self, the more strongly a change in that role would impact one's self-authenticity. Furthermore, Studies 3 and 4 examine coping strategies and find that the effect of role change on inauthenticity is attenuated when people focus on the present.

Across studies, we empirically show that COVID-19-related role changes increase inauthenticity, but a present-focused coping strategy can minimize inauthenticity. These effects are demonstrated in samples drawn from both the USA (Studies 1–3) and Hong Kong (Study 4), and using both experiments (Studies 2 and 3) and surveys (Studies 1 and 4). Survey data are important because, while experiments demonstrate causal relationships, the surveys demonstrate the effects without potential experimental artifacts and, hence, establish ecological validity. Moreover, the survey results of Study 4 hint at the possibility that people may not default to the most effective strategy to cope. They are equally likely to focus on the future as they are to focus on the present under COVID-19, despite present-focused coping being a more effective strategy (per Study 3). Absent effective coping responses, changes to social roles (even if subtle and imposed by external factors) can reduce people's long-term psychological wellbeing.

## Contributions and implications

Our research contributes to the growing literature on the psychological effect of COVID-19. While the majority of such research has examined the emotional distress engendered (e.g., [45, 46]), fewer have examined effects on individuals' sense of self [47–49] and social identities [7, 8]. We explicitly investigate how COVID-19's disruptions to social systems and social roles can undermine self-authenticity. Self-authenticity is essential to health and wellbeing and experiences of inauthenticity can become chronic over time [10, 12]. Thus, our findings extend the literature by documenting another important, and potentially lasting, psychological consequence of COVID-19.

Moreover, we contribute to the authenticity literature by identifying a novel and timely antecedent. Prior research related to social roles and authenticity shows that lack of integration across social roles (e.g., [11, 22]) and adoption of different roles [23] can elicit inauthenticity. Here, we find that disruptions to existing roles also give rise to inauthenticity, even when these disruptions are attributed to an uncontrollable external shock (i.e., COVID-19). Social role disruptions affect inauthenticity when (and hence, because) the disrupted roles are central to people's sense of self. Moreover, the effects of social role disruption on inauthenticity occur even when controlling for valence—that is, (i) whether COVID-19's impact on a particular role was perceived positively or negatively (Study 2) or (ii) whether the COVID-19 related role changes themselves were perceived positively or negatively (Studies 3 and 4). This is important because it suggests that the observed effect is not driven by perceived negativity of COVID-19 or perceived negativity of changes, per se. Indeed, role changes need not be perceived negatively. [Study 2 found a positive correlation between the extent of role change and valence of change ($r = .23$, $p < .001$), but Study 3 found a negative correlation ($r = -.17$, $p = .003$)]. These results have implications beyond COVID-19. A wide variety of role transitions (e.g., parenthood, graduation, retirement) might induce inauthenticity, as might changes to other aspects of life important to one's sense of self (e.g., personal hobbies, lifestyle choices).

Additionally, we contribute to the literature on temporal-focused coping. Past research related to temporal-focused coping in crises suggest that the effectiveness of different temporal foci depends on context. Past-focused coping is effective when the end of a crisis brings a return to normalcy. For example, spinal cord injury patients experience less psychological distress if they focus on "getting back to normality" [50]. Future-focused coping is effective when people are able to plan. For example, Americans traumatized by the 9/11 attack coped best when they predicted and planned their lives [51]. An ongoing pandemic like COVID-19 shows some parallels to other crises, such as homelessness or unemployment, where people may never return to normalcy and cannot predict their new normal. In such insecure contexts, people cope best with a present focus [38–40]. Building on the temporal-focused coping literature, we find that present-focus facilitates coping with inauthenticity under COVID-19. Specifically, people's tendency to be present-focused (but not past-focused or future-focused) moderates COVID-19's effect on authenticity, such that maintaining a present-focus insulates people from the negative effects of social role change on self-authenticity (Study 4). Moreover, pit against each other, present-focused coping outperforms both past- and future-focused coping (Study 3). It is, however, possible that either past- or future-focus coping may become more useful as the pandemic progresses, assuming that normalcy gradually returns and people are better able to plan for the future. Future research can explore that possibility. In the meantime, our results support the efficacy of present-focused coping while COVID-19 remains an ongoing pandemic.

These results have practical implications for temporal-focus coping. For example, public health efforts should encourage thinking about the here-and-now and coping "one day at a

time." Relatedly, practitioners may encourage mindfulness practice, which prompts unjudgmental awareness of the present [52]. Indeed, mindfulness is correlated with self-authenticity [53], and mindfulness alleviates anxiety and depression under COVID-19 [54]. Exploratory analyses of Study 3 hint that present-focus coping is even more effective if people chronically practice mindfulness (S5 Appendix in S1 File) but additional research is required on this issue.

Our findings also contribute to literature on coping with self-inauthenticity. Our findings suggest that previously established mechanisms for coping with inauthenticity (i.e., thinking about the past; [15, 41]) are less effective during this ongoing pandemic. We speculate this is because during the ongoing pandemic, where changes are underway and the return to normalcy is unforeseeable, thinking about the past may, in fact, make salient discontinuity of one's roles (as opposed to increasing continuity as shown in prior research; e.g., [15, 41]). The departure of our strategy from prior research suggests that the optimal temporal strategies to cope with inauthenticity may be context specific.

## Limitations and future directions

We conjecture that present-focused coping works by making role discontinuity less salient. The exact mechanism underlying this coping process warrants further research. One possibility is that role discontinuity under COVID-19 is related to uncertainty and external locus of control, which triggers avoidant coping (e.g., denial and disengagement; [55, 56]); avoidant coping styles have furthermore been linked to inauthenticity [9]. But by encouraging a present focus, avoidant coping is reduced (see exploratory analysis in S5 Appendix in S1 File). Rather than prompting avoidance, a present focus may instead prompt adaptation and potentially self-growth. Future research can examine how temporal focus affects self-authenticity in role transitions and adaptions. Such research may be key for understanding the conditions that might cause temporary states of inauthenticity, such as those experienced under COVID-19, to develop into chronic inauthenticity over time.

As previously alluded to, the experience of social role change under COVID-19 can be thought of as a liminal period. Liminality research suggests that people may become more authentically "themselves" through the process of liminality if they construe changes in identity as a growth process [57]. Relatedly, identity construction during liminal periods (e.g., during online dating) may reduce the discrepancy between actual-self and ideal-self [58], and the reduced self-discrepancy is associated with feeling authentic [10]. Thus, whereas a maladaptive coping strategy could induce chronic inauthenticity, identity work that emphasizes growth could enhance authenticity in the long run. These findings raise the possibility of a silver lining to disruptions under COVID-19: changes such working from home or reducing travels may give people the opportunity and motivation to explore themselves, develop skills, and pursue their goals, which may, in turn, allow people to work towards their ideal selves.

Moreover, COVID-19 might be associated with temporal landmarks, which are events that stand in marked contrast to ordinary occurrences [59–61]. Temporal landmarks help people psychologically separate themselves from an undesirable past by creating a sense of "fresh start" and motivating pursuits of new goals [59, 60]. This enables people to better organize their present and immediate future by setting and pursuing new goals. COVID-19 might present temporal landmarks in a few different ways. First, the onset of COVID-19 could be seen as a temporal landmark denoting a new time period. If so, COVID-19 might offer an opportunity for people to psychologically dissociate from their past, imperfect selves, and strengthen their intentions to pursue their goals [59, 60]. Second, as the pandemic progresses, temporal landmarks might be created by changes in government measures (e.g., end of a national lockdown) and individual circumstances (e.g., first day back to work). These temporal landmarks could

represent a return to normalcy and thus may help people separate themselves from the self under COVID-19, an inauthentic and liminal self that existed earlier in the pandemic. Longitudinal research thus could study the long-term impact of COVID-19 on self-authenticity, particularly in relation to the intriguing possibility that COVID-19 might, in ways, help people gain a new authentic self.

Finally, our results suggest that social role disruption causes inauthenticity by undermining self-continuity, and we propose present-focus as a way to cope with this inauthenticity. However, besides self-continuity, there are other psychological mechanisms that can affect authenticity. One such mechanism is personal autonomy, which relates to the freedom of full expression and choice that gives rise to authenticity [11, 12]. COVID-19 measures impose constraints on people's choice and behaviors. For example, adult children who are eager to visit and care for their ill parents may be restricted from doing so; students who study oversea may be unable to return home for important occasions. People may thus find it difficult to act with a full sense of self-expression and choice in their social roles. Indeed, past research related to authenticity under COVID-19 does so by way of autonomy, specifically, autonomy in the workplace. Anicich et al. [47] find that COVID-19 weakens employees' sense of autonomy ("I feel like I am able to truly be myself right now"), while Dobson [62] finds that working from home in COVID-19 enhances workers' autonomy and authenticity. One way to interpret these results is that preserving autonomy can safeguard authenticity. However, to the extent that maintaining autonomy in one social role may cause disruption to another social role, one's overall sense of authenticity may still be reduced. It is therefore important for future research to look at systems of social roles rather than examining social roles in isolation.

## Conclusion

In sum, various COVID-19 protective measures safeguard physical health, but they cause disruptions in people's social roles. This can take a toll on people's sense of authenticity, particularly if people's coping responses are suboptimal. The present research provides initial evidence that levels of inauthenticity increase as the social roles in people's lives are upended by COVID-19, but this effect can be attenuated by focusing on the here-and-now. These findings contribute to the literature in several ways. Firstly, our findings extend the literature on COVID-19's impact on mental health by documenting another important and potentially long-lasting psychological consequence: self-inauthenticity. Secondly, we identify social role disruption as an antecedent to self-inauthenticity, and thus contribute to the authenticity literature. Lastly, our findings contribute to the literature on how to cope with COVID-19 and its associated social role disruptions, as well as the literature on how to cope with self-inauthenticity. We demonstrate that adopting a present-focus, as opposed to a past- or future-focus, attenuates the effect of social role disruptions on self-inauthenticity.

## Supporting information

**S1 File.**
(DOCX)

## Author Contributions

**Conceptualization:** Jingshi (Joyce) Liu, Amy N. Dalton.

**Formal analysis:** Jingshi (Joyce) Liu.

**Funding acquisition:** Amy N. Dalton.

**Investigation:** Jingshi (Joyce) Liu, Jeremy Lee.

**Methodology:** Jingshi (Joyce) Liu, Amy N. Dalton, Jeremy Lee.

**Supervision:** Amy N. Dalton.

**Writing – original draft:** Jingshi (Joyce) Liu.

**Writing – review & editing:** Jingshi (Joyce) Liu, Amy N. Dalton, Jeremy Lee.

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
