## [Decision Letter · Decision Letter 0]

20 Jul 2021

PONE-D-21-18664

The self under COVID-19: Social role disruptions, self-authenticity and present-focused coping

PLOS ONE

Dear Dr. Lee,

Thank you for submitting your manuscript to PLOS ONE. After careful consideration, we feel that it has merit but does not fully meet PLOS ONE’s publication criteria as it currently stands. Therefore, we invite you to submit a revised version of the manuscript that addresses the points raised during the review process.

We look forward to receiving your revised manuscript.

Kind regards,

Frantisek Sudzina

Academic Editor

PLOS ONE

Journal Requirements:

Reviewers' comments:

Reviewer's Responses to Questions

**Comments to the Author**

1. Is the manuscript technically sound, and do the data support the conclusions?

Reviewer #1: Yes

Reviewer #2: Yes

2. Has the statistical analysis been performed appropriately and rigorously? 

Reviewer #1: Yes

Reviewer #2: Yes

3. Have the authors made all data underlying the findings in their manuscript fully available?

Reviewer #1: Yes

Reviewer #2: No

4. Is the manuscript presented in an intelligible fashion and written in standard English?

Reviewer #1: Yes

Reviewer #2: Yes

5. Review Comments to the Author

Reviewer #1: 1.The manuscript has good identification of keywords and its cognitive appeal to readers' knowledge, skills and motives turning effective the communication. In terms of data, the operability of the hypotheses is relevant and leads to an understanding of the meaning of the results which are supported by them, the use of experiences supported by surveys anticipates the doubts that have arisen and increase the rigour of the study which, however, should be much more explained in words by the numerical results as it expands the possibility of increasing the number of readers and their interest. Here is the suggestion. As for sampling, the intention to represent the study and its consequences seem to us to be sufficient to identify the effects arising from the experiences and other conditions imposed on the investigation namely in the issue of changing roles so profusely focused and replicated through the effects of COVID-19.The conclusions were credibly drawn and show adequate testing of hypotheses with significant results.

2. The statistical method is appropriate and well conducted, translating a good analysis of the data collected in a detailed way and with good information about the acceptability of the confirmed hypotheses.

3. The authors made all data underlying the findings in their manuscript fully available.

4. The work expresses itself in appropriated English, although with some vocabulary gaps but tiny and that go unnoticed.

Reviewer #2: My biggest concern is about unavailable data.

I cannot agree with the generalization conclusion, not even for both "countries" students

Humans are by nature social animals since Aristoteles and in the context of the paper I totally agree but nowadays this is not a consensual statement (eg. Thomas Hobbes). It’s not important but could be rephrased.

Also the theoretical support about “present-focused coping” could be more updated.

(e.g. Time Perspective Theory; Review, Research and Application Essays in Honor of Philip G. Zimbardo, 2015).

I really appreciate the conclusions.

6. PLOS authors have the option to publish the peer review history of their article (what does this mean?). If published, this will include your full peer review and any attached files.

Reviewer #1: No

Reviewer #2: No

---

## [Author Response · Author response to Decision Letter 0]

13 Aug 2021

Dear Dr. Sudzina, 

We begin by extending our thanks for the opportunity to revise our manuscript, “The self under COVID-19: Social role disruptions, self-authenticity and present-focused coping.” Per your invitation, we revised the paper based on the feedback of the review team. Our revision efforts were aimed directly at remedying the shortcomings identified. Thanks to the review team’s stewardship, the manuscript is clearer conceptually and empirically. 

For completeness, we address in detail each of the points below. 

Academic Editor: 

Per the journal requirements, we confirm that (1) our reference list does not contain any retracted articles, (2) our manuscript’s formatting adheres to PLOS ONE’s style guide, and (3) our ethics statement appears in the Method section of our manuscript. 

In addition to addressing the concerns of the reviewers, we took the liberty of making a few additional changes that are only editorial and have no bearing on our theory or results. In particular, we updated the examples of COVID-19-related social disruptions in our opening paragraph. We also updated our literature review so that our terminology maps on precisely to that used by the authors we are citing.

Reviewer 1:

Thank you for the integrative review as well as for your support of our work. Your sole recommendation, based on our understanding of your feedback, is that the studies “should be much more explained in words by the numerical results as it expands the possibility of increasing the number of readers and their interest.” 

We took this recommendation to heart and focused our efforts on allaying it. All studies now explain in greater detail the predictions, rationale for each analysis, and the implications of the results. We have also clarified parts of the study introductions (mostly to reinforce our predictions) and methods (mostly to make explicit our IVs and DVs). We thank you for pointing out this issue, and we believe the readability of the manuscript is stronger thanks to these additions. 

Our changes have added several additional lines of text throughout the document, primarily to the study descriptions. We hope these additions are within the parameters and warranted by the length-to-contribution ratio.

Reviewer 2:

We are grateful for the guidance offered and for your appreciation of our findings. We believe our revision has fully addressed each of your recommendations, which are fourfold. 

(1) “My biggest concern is about unavailable data” 

All data can be accessed from a downloadable Dropbox folder. On our initial submission, we made that link available in the methods sections of studies 2 and 4. The link is somewhat buried in these locations so we have now added it to a more prominent location, at the end of the section called “Sample size, statistical analyses and procedure disclosure.” All the data will continue to be available through this link. 

(2) “I cannot agree with the generalization conclusion, not even for both "countries’ students” 

We have toned down the claim that replication in different populations implies generalizability. We now state concretely (and only) that the effect of role change and the benefit of present-focused coping were observed in both the USA sample and Hong Kong sample.

This change is reflected in text in the “Abstract”, the “Discussion” of Study 4, and the “Study Summary” under the “General Discussion.” 

(3) “Humans are by nature social animals since Aristoteles and in the context of the paper I totally agree but nowadays this is not a consensual statement (eg. Thomas Hobbes). It’s not important but could be rephrased.”

We have removed the claim that “humans are by nature social animals” and replaced it with the statement that “Social roles are fundamental to people’s sense of self.”

This change is reflected in the first sentence of the “Introduction.”

(4) “Also the theoretical support about “present-focused coping” could be more updated.

(e.g. Time Perspective Theory; Review, Research and Application Essays in Honor of Philip G. Zimbardo, 2015).”

We thank you for providing us with this reference and have incorporated your feedback in two ways. 

First, in the section “Coping with inauthenticity under COVID-19: the role of temporal perspectives,” we now include a clear definition of temporal perspective and briefly review relevant literature on temporal perspective to acknowledge recent developments in the literature. 

Second, in the section “Contributions and Implications,” we now reference an additional recent article hinting at the potential adaptiveness of maintaining a present focus in socially insecure contexts. We stopped short of drawing a direct parallel between the papers we cite and our own research given the different social contexts being studied and dependent variables being measured but we do note that all are relevant to the underlying point that present-focused coping can be an adaptive strategy during crises. 

We thank the team for showing us a clear path to publication and hope the details of our revision notes met your expectations. 

Kind regards, 

Jingshi (Joyce) Liu

Amy N. Dalton 

Jeremy Lee

---

## [Editor Report · Decision Letter 1]

19 Aug 2021

The “self” under COVID-19: Social role disruptions, self-authenticity and present-focused coping

PONE-D-21-18664R1

Dear Dr. Lee,

We’re pleased to inform you that your manuscript has been judged scientifically suitable for publication and will be formally accepted for publication once it meets all outstanding technical requirements.

Kind regards,

Frantisek Sudzina

Academic Editor

PLOS ONE
---

## [Editor Report · Acceptance letter]

27 Aug 2021

PONE-D-21-18664R1 

The “Self” under COVID-19:
Social role disruptions, self-authenticity and present-focused coping 

Dear Dr. Lee:

I'm pleased to inform you that your manuscript has been deemed suitable for publication in PLOS ONE. Congratulations! Your manuscript is now with our production department. 

Kind regards, 

on behalf of

Dr. Frantisek Sudzina 

Academic Editor

PLOS ONE